# Logarithmic landscape and power-law escape rate of SGD

## Abstract

Stochastic gradient descent (SGD) undergoes complicated multiplicative noise for the mean-square loss. We use this property of the SGD noise to derive a stochastic differential equation (SDE) with simpler additive noise by performing a random time change. In the SDE, the loss gradient is replaced by the logarithmized loss gradient. By using this formalism, we obtain the escape rate formula from a local minimum, which is determined not by the loss barrier height $\Delta L = L(\theta^s) - L(\theta^*)$ between a minimum $\theta^*$ and a saddle $\theta^s$ but by the logarithmized loss barrier height $\Delta \log L = \log[L(\theta^s)/L(\theta^*)]$. Our escape-rate formula strongly depends on the typical magnitude $h^*$ and the number $n$ of the outlier eigenvalues of the Hessian. This result explains an empirical fact that SGD prefers flat minima with low effective dimensions, which gives an insight into implicit biases of SGD.

## 1 Introduction

Deep learning has achieved breakthroughs in various applications in artificial intelligence such as image classification (Krizhevsky et al., 2012; LeCun et al., 2015), speech recognition (Hinton et al., 2012), natural language processing (Collobert & Weston, 2008), and natural sciences (Iten et al., 2020; Bapst et al., 2020; Seif et al., 2021). Such unparalleled success of deep learning hinges crucially on stochastic gradient descent (SGD) or its variants as an efficient training algorithm.

Although the loss landscape is highly nonconvex, the SGD often succeeds in finding a global minimum. It has been argued that the SGD noise plays a key role in escaping from local minima (Jastrzębski et al., 2017; Wu et al., 2018; 2020; Zhu et al., 2019; Meng et al., 2020; Xie et al., 2021; Liu et al., 2021). It has also been suggested that SGD has an implicit bias that is beneficial for generalization. That is, SGD may help the network to find *flat minima*, which are considered to imply good generalization (Keskar et al., 2017; Hoffer et al., 2017; Wu et al., 2018). How and why does SGD help the network escape from bad local minima and find flat minima? These questions have been addressed in several works, and it is now recognized that the SGD noise strength and structure importantly affect the efficiency of escape from local minima. Our work follows this line of research, and add new theoretical perspectives.

In physics and chemistry, escape from a local minimum of the (free) energy landscape due to thermal noise at temperature $T$ has been thoroughly discussed (Kramers, 1940; Langer, 1969). When the (free) energy barrier is given by $\Delta E$, the escape rate is proportional to $e^{-\Delta E/T}$, which is known as the Arrhenius law. By analogy, in machine learning, escape from a local minimum of the loss function is considered to be determined by the loss barrier height $\Delta L = L(\theta^s) - L(\theta^*)$, where $L(\theta)$ denotes the loss function at the network parameters $\theta$, $\theta^*$ stands for a local minimum of $L(\theta)$, and $\theta^s$ denotes a saddle point that separates $\theta^*$ from other minima. If we assume that the SGD noise is uniform and isotropic, which is often assumed in machine-learning literature (Jastrzębski et al., 2017), the escape rate is proportional to $e^{-\Delta L/D}$, where $D$ denotes the SGD noise strength.

In this paper, we show that inhomogeneity of the SGD noise strength brings about drastic modification for the mean-square loss. It turns out that the escape rate is determined by the *logarithmized* loss barrier height $\Delta \log L = \log L(\theta^s) - \log L(\theta^*) = \log[L(\theta^s)/L(\theta^*)]$. In other words, the escape rate is determined by not the difference but the ratio of $L(\theta^s)$ and $L(\theta^*)$. This result means that even if the loss barrier height $\Delta L$ is the same, minima with smaller values of $L(\theta^*)$ are more stable.

Moreover, given the fact that the eigenvalue spectrum of the Hessian at a minimum consists of a bulk of almost zero eigenvalues and outliers (Sagun et al., 2017; Papyan, 2019), our escape-rate formula implies that *SGD prefers flat minima with a low effective dimension*, where the effective dimension is defined as the number of outliers (MacKay, 1992) and flatness is measured by a typical magnitude of outlier eigenvalues (Keskar et al., 2017). The previous theories (Jastrzębski et al., 2017; Wu et al., 2018; Zhu et al., 2019; Meng et al., 2020; Xie et al., 2021; Liu et al., 2021) have also successfully explained that SGD prefers flat minima, but not shown the preference of small effective dimensions. The logarithmized loss naturally explains the latter, and sheds light on implicit biases of SGD.

**Main contributions**: We obtain the following main results:

- We derive an equation for approximating the SGD noise in Eq. (6). Remarkably, the SGD noise strength in the mean-square loss is shown to be proportional to the loss function, which is experimentally verified in Sec. 5.2. A key ingredient in deriving Eq. (6) is the *decoupling approximation* given in Eq. (7). This is a novel approximate method introduced in our analysis, and hence we experimentally verify it in Sec. 5.1.

- We derive a novel stochastic differential equation (SDE) in Eq. (14) via a random time change introduced in Eq. (13). Although the original SDE (4) has a multiplicative noise, the transformed SDE (14) has a simple additive noise with the gradient of the logarithmic loss. This shows the convenience of the logarithmic loss landscape for understanding SGD.

- We derive a novel form of SGD escape rate from a local minimum in Eq. (16). Remarkably, the escape rate depends on the ratio between $L(\theta^*)$ and $L(\theta^s)$. In Sec. 5.3, we experimentally test the validity of this result for linear regressions.

- Our escape rate crucially depends on the flatness and the effective dimension, which shows that SGD has implicit biases towards flat minima with low effective dimension. We also show in Eq. (18) that a local minimum with an effective dimension $n$ greater than a certain critical value $n_c$ becomes unstable.

**Related works**: The role of the SGD noise structure has been discussed in some previous works (Zhu et al., 2019; Xie et al., 2021; Liu et al., 2021; Meng et al., 2020; Wojtowytsch, 2021). It was pointed out that the anisotropic nature of the SGD noise is important: the SGD noise covariance matrix is aligned with the Hessian of the loss function, which is beneficial for escape from sharp minima (Zhu et al., 2019; Xie et al., 2021; Liu et al., 2021). These previous works, however, do not take the inhomogeneity of the SGD noise strength into account, and consequently, escape rates derived there depend exponentially on the loss barrier height, which differs from our formula.

Compared with the anisotropy of the SGD noise, the inhomogeneity of the SGD noise strength has been less explored. In (Meng et al., 2020; Wojtowytsch, 2021), the SGD dynamics under a state-dependent noise is discussed. However, in these previous works, the connection between the noise strength and the loss function was not theoretically established, and the logarithmized loss landscape was not discussed. The instability due to large effective dimensions was also not shown. Another recent work (Pesme et al., 2021) observed that the noise is proportional to the loss for specific simple models. In our paper, such a result is derived for more generic models. Gürbüzbalaban et al. (2021) showed that SGD will converge to a heavy-tailed stationary distribution due to a multiplicative nature of the SGD noise in a simple linear regression problem. Our paper strengthens this result: we argue that such a heavy-tailed distribution generically appears for the mean-square loss.

## 2 BACKGROUND

### 2.1 SETUP

We consider supervised learning. Let $\mathcal{D} = \{(x^{(\mu)}, y^{(\mu)}) : \mu = 1, 2, \ldots, N\}$ be the training dataset, where $x^{(\mu)} \in \mathbb{R}^d$ denotes a data vector and $y^{(\mu)} \in \mathbb{R}$ be its label. The network output for a given input $x$ is denoted by $f(\theta, x) \in \mathbb{R}$, where $\theta \in \mathbb{R}^P$ stands for a set of trainable parameters with $P$ being the number of trainable parameters (Extension to the multi-dimensional label and output is straightforward). In this work, we focus on the mean-square loss

$$L(\theta) = \frac{1}{2N} \sum_{\mu=1}^{N} \left[ f(\theta, x^{(\mu)}) - y^{(\mu)} \right]^2 =: \frac{1}{N} \sum_{\mu=1}^{N} \ell_\mu(\theta). \tag{1}$$

The training proceeds through optimization of $L(\theta)$. In most machine-learning applications, the optimization is done via SGD or its variants. In SGD, the parameter $\theta_{k+1}$ at the time step $k+1$ is determined by

$$\theta_{k+1} = \theta_k - \eta \nabla L_{B_k}(\theta_k), \quad L_{B_k}(\theta) = \frac{1}{2B} \sum_{\mu \in B_k} \ell_\mu(\theta), \tag{2}$$

where $\eta > 0$ is the learning rate, $B_k \subset \{1, 2, \dots, N\}$ with $|B_k| = B$ is a mini-batch used at the $k$th time step, and $L_{B_k}$ denotes the mini-batch loss. Since the training dataset $\mathcal{D}$ is randomly divided into mini-batches, the dynamics defined by Eq. (2) is stochastic. When $B = N$, the full training data samples are used for every iteration. In this case, the dynamics is deterministic and called gradient descent (GD). SGD is interpreted as GD with stochastic noise. By introducing the SGD noise $\xi_k = -[\nabla L_{B_k}(\theta_k) - \nabla L(\theta_k)]$, Eq. (2) is rewritten as

$$\theta_{k+1} = \theta_k - \eta \nabla L(\theta_k) + \eta \xi_k. \tag{3}$$

Obviously, $\langle \xi_k \rangle = 0$, where the brackets denote the average over possible choices of mini-batches. The noise covariance matrix is defined as $\Sigma(\theta_k) := \langle \xi_k \xi_k^{\mathrm{T}} \rangle$. The covariance structure of the SGD noise is important in analyzing the SGD dynamics, which will be discussed in Sec. 3.1.

## 2.2 STOCHASTIC DIFFERENTIAL EQUATION FOR SGD

When the parameter update for each iteration is small, which is typically the case when the learning rate $\eta$ is small enough, we can consider the continuous-time approximation (Li et al., 2017; Smith & Le, 2018). By introducing a continuous time variable $t \in \mathbb{R}$ and regarding $\eta$ as an infinitesimal time step $dt$, we have a SDE

$$d\theta_t = -\nabla L(\theta_t)dt + \sqrt{\eta \Sigma(\theta_t)} \cdot dW_t, \tag{4}$$

where $dW_t \sim \mathcal{N}(0, I_P dt)$ with $I_n$ being the $n$-by-$n$ identity matrix, and the multiplicative noise $\sqrt{\eta \Sigma(\theta_t)} \cdot dW_t$ is interpreted as Itô since the noise $\xi_k$ in Eq. (3) depends on $\theta_k$ but not on $\theta_{k+1}$. Throughout this work, we consider the continuous-time approximation (4) with Gaussian noise.

In machine learning, the gradient Langevin dynamics (GLD) is also considered, in which the isotropic and uniform Gaussian noise is injected into the GD as

$$d\theta_t = -\nabla L(\theta_t)dt + \sqrt{2D}dW_t, \tag{5}$$

where $D > 0$ corresponds to the noise strength (it is also called the diffusion coefficient) (Sato & Nakagawa, 2014; Zhang et al., 2017b; Zhu et al., 2019). The stationary probability distribution $P_{\mathrm{GLD}}(\theta)$ of $\theta$ for GLD is given by the Gibbs distribution $P_{\mathrm{GLD}}(\theta) \propto e^{-L(\theta)/D}$. We will see in Sec. 4 that the SGD noise structure, which is characterized by $\Sigma(\theta)$, drastically alters the stationary distribution and the escape rate from a local minimum.

## 3 THEORETICAL FORMULATION

### 3.1 STRUCTURE OF THE SGD NOISE COVARIANCE

The SGD noise covariance matrix $\Sigma(\theta)$ significantly affects the dynamics (Jastrzębski et al., 2017; Smith & Le, 2018; Zhu et al., 2019; Ziyin et al., 2021). In this section, under some approximations, we derive the following expression of $\Sigma(\theta)$ for the mean-square loss within the the basin of attractions (or the "valley") of a local minimum $\theta^*$:

$$\Sigma(\theta) \approx \frac{2L(\theta)}{NB} H(\theta^*), \tag{6}$$

where $H(\theta) = \nabla^2 L(\theta)$ is the Hessian. It should be noted that $\nabla L(\theta^*) = 0$ and $H(\theta^*)$ is positive semidefinite at any local minima $\theta^*$. We give a derivation below and the list of the approximations and their justifications in Appendix A. In particular, the *decoupling approximation* is a new tool and plays a key role in the derivation. It states that the quantities $\ell_\mu$ and $C_f^{(\mu)}(\theta) := \nabla f(\theta, x^{(\mu)}) \nabla f(\theta, x^{(\mu)})^{\mathrm{T}}$ are uncorrelated, which implies

$$\frac{1}{N} \sum_{\mu=1}^{N} \ell_\mu C_f^{(\mu)}(\theta) \approx \left( \frac{1}{N} \sum_{\mu=1}^{N} \ell_\mu \right) \cdot \left( \frac{1}{N} \sum_{\mu=1}^{N} C_f^{(\mu)} \right) = L(\theta) \frac{1}{N} \sum_{\mu=1}^{N} C_f^{(\mu)}(\theta). \tag{7}$$

This approximation is promising for large networks in which $\nabla f$ looks a random vector. In Sec. 5.1, we experimentally verify the decoupling approximation for the entire training dynamics.

Our formula (6) possesses two important properties. First, the noise is aligned with the Hessian, which has been well known and pointed out in the literature (Jastrzębski et al., 2017; Zhu et al., 2019; Xie et al., 2021; Liu et al., 2021). If the loss landscape has flat directions, which correspond to the directions of the Hessian eigenvectors belonging to vanishingly small eigenvalues, the SGD noise does not work along those directions. Consequently, SGD dynamics is frozen along those flat directions, which effectively reduces the dimension of the parameter space explored by SGD dynamics. This plays an important role in the escape efficiency. Indeed, we will see that the escape rate crucially depends on the effective dimension of a given local minimum.

Second, the noise is proportional to the loss function, which is indeed experimentally confirmed in Sec. 5.2. This property has not been pointed out and not been taken into account in previous studies (Jastrzębski et al., 2017; Zhu et al., 2019; Xie et al., 2021; Liu et al., 2021) and therefore gives new insights into the SGD dynamics. Indeed, this property allows us to formulate the Langevin equation on the logarithmized loss landscape with simple additive noise as discussed in Sec. 3.2. This new formalism yields the power-law escape rate, i.e. Eqs. (15) and (16), and the importance of the effective dimension of local minima for their stability.

**Derivation of Eq. (6)**: We start from an analytic expression of $\Sigma(\theta)$, which reads

$$\Sigma(\theta) = \frac{1}{B} \frac{N-B}{N-1} \left( \frac{1}{N} \sum_{\mu=1}^{N} \nabla \ell_\mu \nabla \ell_\mu^{\mathrm{T}} - \nabla L \nabla L^{\mathrm{T}} \right) \simeq \frac{1}{B} \left( \frac{1}{N} \sum_{\mu=1}^{N} \nabla \ell_\mu \nabla \ell_\mu^{\mathrm{T}} - \nabla L \nabla L^{\mathrm{T}} \right), \quad (8)$$

where $B \ll N$ is assumed in the second equality. The derivation of Eq. (8) is found in Jastrzębski et al. (2017); Smith & Le (2018). Usually, the gradient noise variance dominates the square of the gradient noise mean, and hence the term $\nabla L \nabla L^{\mathrm{T}}$ in Eq. (8) is negligible.

For the mean-square loss, we have $\nabla \ell_\mu = [f(\theta, x^{(\mu)}) - y^{(\mu)}] \nabla f(\theta, x^{(\mu)})$, and hence

$$\Sigma(\theta) \approx \frac{2}{BN} \sum_{\mu=1}^{N} \ell_\mu \nabla f(\theta, x^{(\mu)}) \nabla f(\theta, x^{(\mu)})^{\mathrm{T}} = \frac{2}{BN} \sum_{\mu=1}^{N} \ell_\mu C_f^{(\mu)}(\theta) \approx \frac{2L(\theta)}{NB} \sum_{\mu=1}^{N} C_f^{(\mu)}(\theta), \quad (9)$$

where the decoupling approximation (7) is used in the last equality. Equation (9) is directly related to the Hessian of the loss function near a (local or global) minimum. The Hessian is written as

$$H(\theta) = \nabla^2 L(\theta) = \frac{1}{N} \sum_{\mu=1}^{N} C_f^{(\mu)}(\theta) + \frac{1}{N} \sum_{\mu=1}^{N} \left[ f(\theta, x^{(\mu)}) - y^{(\mu)} \right] \nabla^2 f(\theta, x^{(\mu)}). \quad (10)$$

It is shown by Papyan (2018) that the last term of Eq. (10) does not contribute to outliers (i.e. large eigenvalues) of the Hessian. Dynamics near a local minimum is governed by outliers, and hence we can ignore this term. At $\theta = \theta^*$, we therefore obtain

$$H(\theta^*) \approx \frac{1}{N} \sum_{\mu=1}^{N} C_f^{(\mu)}(\theta^*). \quad (11)$$

Let us assume $C_f^{(\mu)}(\theta) \approx C_f^{(\mu)}(\theta^*)$ for $\theta$ within the valley of a local minimum $\theta^*$. We then obtain the desired expression (6) by substituting it into Eq. (9).

## 3.2 LOGARITHMIZED LOSS LANDSCAPE

Let us consider the Itô SDE (4) with the SGD noise covariance (6) near a local minimum $\theta^*$, which is written as

$$d\theta_t = -\nabla L(\theta_t) dt + \sqrt{\frac{2\eta L(\theta_t)}{B} H(\theta^*)} \cdot dW_t. \quad (12)$$

Let us consider a stochastic time $t(\tau)$ for $\tau \geq 0$ as

$$\tau = \int_0^{t(\tau)} dt' \, L(\theta_{t'}), \quad (13)$$

and perform a random time change from $t$ to $\tau$ (Øksendal, 1998). Correspondingly, we introduce the Wiener process $d\tilde{W}_\tau \sim \mathcal{N}(0, I_P d\tau)$. Since $d\tau = L(\theta_t)dt$, we have $d\tilde{W}_\tau = \sqrt{L(\theta_t)} \cdot dW_t$. In terms of the notation $\tilde{\theta}_\tau = \theta_t$, Eq. (12) is expressed as

$$d\tilde{\theta}_\tau = -\frac{1}{L(\tilde{\theta}_\tau)}\nabla L(\tilde{\theta}_\tau)d\tau + \sqrt{\frac{2\eta H(\theta^*)}{B}}d\tilde{W}_\tau = -\left[\nabla \log L(\tilde{\theta}_\tau)\right]d\tau + \sqrt{\frac{2\eta H(\theta^*)}{B}}d\tilde{W}_\tau. \quad (14)$$

We should note that at a global minimum with $L(\theta) = 0$, which is realized in an overparameterized regime (Zhang et al., 2017a), the random time change through Eq. (13) is ill-defined since $\tau$ is frozen at a finite value once the model reaches a global minimum. We can overcome this difficulty by adding an infinitesimal constant $\epsilon > 0$ to the loss, which makes the loss function positive without changing the finite-time dynamics like the escape from a local minimum $\theta^*$ with $L(\theta^*) > 0$.

In this way, the Langevin equation on the loss landscape $L(\theta)$ with multiplicative noise is transformed to that on the logarithmic loss landscape $U(\theta) = \log L(\theta)$ with simpler additive noise. This formulation indicates the importance of considering the logarithmized loss landscape $U(\theta) = \log L(\theta)$. In the following, we use Eq. (14) to discuss the escape efficiency from local minima.

## 4    ESCAPE RATE FROM LOCAL MINIMA

Let us consider a local minimum $\theta^*$ and its basin of attraction $\mathcal{A}_{\theta^*}$, which is the set of all the starting points $\theta_0$ such that $\theta_t$ tends to $\theta^*$ as $t \to \infty$ if there is no noise. In order to escape from the basin of $\theta^*$, $\theta_t$ must go through a saddle $\theta^s$ with the help of noise. The escape time is defined as $\tau = \inf\{t > 0 : \theta_t \notin \mathcal{A}_{\theta^*}, \theta_0 = \theta^*\}$. The escape rate $\kappa$ is then defined as the inverse of the mean escape time: $\kappa = \langle\tau\rangle^{-1}$. Our idea for evaluating the escape rate is applying the Kramers formula (Kramers, 1940) or its high-dimensional generalization (Langer, 1969; Bovier et al., 2004; Berglund, 2013) to Eq. (14). It should be noted that the Kramers formula is applicable only for additive noise, and hence it cannot be used for the original SDE (4) with multiplicative noise.

First, we present main results [Eqs. (15), (16), and (17)], and give their derivations later. For $P = 1$ (i.e. the single-variable case $\theta \in \mathbb{R}$), the escape rate through a saddle $\theta^s$ is given by

$$\kappa = \frac{1}{2\pi}\sqrt{h^*|h^s|}\left[\frac{L(\theta^s)}{L(\theta^*)}\right]^{-\left(\frac{1}{2}+\frac{B}{\eta h^*}\right)}, \quad (15)$$

where $h^* = H(\theta^*)$. This is a variant of the Kramers formula (Kramers, 1940), which is accurate when $\kappa$ is small. We can derive Eq. (15) from Eq. (14) without any further approximations.

For $\theta \in \mathbb{R}^P$ with $P > 1$, the analysis is more involved and needs some assumptions and approximations, which are listed in Appendix A. We assume that the eigenvalue spectrum of the Hessian $H(\theta^*)$ at the local minimum $\theta^*$ consists of bulk of almost zero eigenvalues and outliers, which is indeed empirically correct (Sagun et al., 2017; Papyan, 2019). We denote by $h^*$ and $n$ a typical magnitude of outlier eigenvalues (i.e. the flatness) and the number of outlier eigenvalues (i.e. the effective dimension), respectively. As we discussed in Sec. 3.1, the SGD dynamics is frozen along flat directions. Therefore SGD dynamics around the local minimum $\theta^*$ is restricted to the $n$-dimensional manifold spanned by the outlier eigenvectors $v_1, v_2, \ldots, v_n \in \mathbb{R}^P$ of $H(\theta^*)$. Now we parameterize $\theta$ by using $n$ parameters $z_1, z_2, \ldots, z_n \in \mathbb{R}$ as $\theta = \theta^* + \sum_{i=1}^n z_i v_i$. The Hessian restricted to this outlier subspace is then written as $\hat{H}(\theta) := \nabla_z^2 L(\theta) \in \mathbb{R}^{n \times n}$.

We then obtain the following escape rate formula for $P > 1$:

$$\kappa = \frac{|h_e^s|}{2\pi}\sqrt{\frac{\det \hat{H}(\theta^*)}{|\det \hat{H}(\theta^s)|}}\left[\frac{L(\theta^s)}{L(\theta^*)}\right]^{-\left(\frac{B}{\eta h^*}+1-\frac{n}{2}\right)}, \quad (16)$$

where $h_e^s$ is the negative eigenvalue of $H(\theta^s)$ corresponding to the escape direction.[1] Again, Eq. (16) is valid when $\kappa$ is small enough. It should be noted that Eq. (16) is reduced to Eq. (15) when $n = 1$.

---

[1] If eigenvalues of $\hat{H}(\theta^*)$ and those of $\hat{H}(\theta^s)$ coincide with each other except for the escape direction $e$, $\det \hat{H}(\theta^*)/\det \hat{H}(\theta^s) = h_e^*/h_e^s$ holds, and Eq. (16) is simplified as $\kappa = (1/2\pi)\sqrt{h_e^*|h_e^s|}[L(\theta^s)/L(\theta^*)]^{-\left(\frac{B}{\eta h^*}+1-\frac{n}{2}\right)}$.

For any $P$, the quasi-stationary distribution $P_{ss}(\theta)$ within the valley of a local minimum $\theta^*$, which can be identified as the stationary distribution restricted to the valley (Bianchi & Gaudillière, 2016), is written as

$$P_{ss}(\theta) \propto L(\theta)^{-\phi}, \quad \phi = 1 + \frac{B}{\eta h^*}. \tag{17}$$

Remarkably, it depends on $L(\theta)$ polynomially rather than exponentially as in the standard GLD (Jastrzębski et al., 2017; Sato & Nakagawa, 2014; Zhang et al., 2017b). This polynomial dependence on $L(\theta)$ is a key feature leading to the escape rate formula.

From Eq. (16) we can obtain some implications. The factor $[L(\theta^s)/L(\theta^*)]^{-\left(\frac{B}{\eta h^*}+1-\frac{n}{2}\right)}$ increases with $h^*$ and $n$, which indicates that sharp minima (i.e. minima with large $h^*$) or minima with large $n$ are unstable. This fact explains why SGD finds flat minima with a low effective dimension $n$. Equation (16) also implies that the effective dimension of any stable minima must satisfy

$$n < n_c := 2\left(\frac{B}{\eta h^*} + 1\right). \tag{18}$$

The instability due to a large effective dimension is a new insight naturally explained by the picture of the logarithmized loss landscape. It arises from the ratio of the determinants of the logarithmized-loss Hessian: $\det \nabla_z^2 U(\theta^*)/|\det \nabla_z^2 U(\theta^s)| = [\det \hat{H}(\theta^*)/|\det \hat{H}(\theta^s)|] \cdot [L(\theta^s)/L(\theta^*)]^{n/2}$.

**Derivation of Eq. (16) and Eq. (17)**: Now we give a derivation of Eqs. (16) and (17). Equation (15) is straightforwardly obtained by putting $P = 1$ in the derivation below (some approximations are introduced below, but all of them are not necessary for $P = 1$).

As we already noted, SGD dynamics is restricted to the $n$-dimensional outlier subspace. First, we assume that the anisotropy of the SGD noise within this $n$-dimensional space is not relevant, and approximate the SGD noise in Eq. (14) as an isotropic one: $\sqrt{2\eta H(\theta^*)/B}d\tilde{W}_\tau \approx \sqrt{2\eta h^*/B}d\tilde{W}_\tau$, where $h^* \in \mathbb{R}^+$ characterizes the magnitude of the Hessian outliers. This assumption is justified when the loss landscape is isotropic within the $n$-dimensional subspace near the minimum. Even if the Hessian at the minimum is not isotropic, this approximation is justified when the directions of the Hessian eigenvectors do not change within the valley. In the latter case, the escape path is a straight line along the direction of a Hessian eigenvector $v_e$, where $e \in \{1, 2, \ldots, n\}$ identifies the escape direction, and $h^*$ corresponds to the Hessian eigenvalue at the minimum along the escape direction, i.e. $h^* = h_e^*$. See Appendix C for the details.

Under this approximation, Eq. (14) becomes $dz_\tau = -\nabla_z U d\tau + \sqrt{2T}d\tilde{W}_\tau$, where $T = \eta h^*/B$. Its quasi-stationary distribution $\tilde{P}_{ss}(\theta)$ within the valley including the local minimum $\theta = \theta^*$ (i.e. $z = 0$) is then given by a Gibbs distribution with respect to $U(\theta)$: $\tilde{P}_{ss}(\theta) \propto e^{-U(\theta)/T} \propto L(\theta)^{-B/(\eta h^*)}$. This is the distribution function of $\tilde{\theta}_\tau$ for a fixed $\tau$. However, what we want is the quasi-stationary distribution of $\theta_t$ for a fixed $t$. In Appendix B, by using Eq. (13), we show that the two distributions are related with each other as $P_{ss}(\theta) \propto L(\theta)^{-1}\tilde{P}_{ss}(\theta)$. We thus obtain Eq. (17).

The escape rate under the Langevin equation with isotropic additive noise is evaluated by using celebrated Kramers formula (Kramers, 1940) or its high-dimensional generalization (Langer, 1969; Bovier et al., 2004; Berglund, 2013). According to it, the escape rate $\kappa_\tau$ *per unit $\tau$* is given by

$$\kappa_\tau = \frac{|u_e^s|}{2\pi}\sqrt{\frac{\det \nabla_z^2 U(\theta^*)}{|\det \nabla_z^2 U(\theta^s)|}}e^{-\Delta U/T}, \tag{19}$$

where $u_e^s$ is the negative eigenvalue of $\nabla_z^2 U(\theta^s)$ corresponding to the escape direction, $\Delta U = U(\theta^s) - U(\theta^*)$ is called the potential barrier, and $\Delta U/T$ is assumed to be large enough. What we really want is the escape rate *per unit time $t$*. It is a reasonable assumption that $\theta_t$ stays close to $\theta^*$ for most times before escape, and hence $\tau$ is approximately given by $\tau \simeq L(\theta^*)t$. The escape rate $\kappa$ per unit $t$ is then given by $\kappa = L(\theta^*)\kappa_\tau$. By combining it with Eq. (19), and substituting $U(\theta) = \log L(\theta)$, we finally obtain Eq. (16).

## 5 EXPERIMENTS

Our key theoretical observation is that the SGD noise strength is proportional to the loss function, which is obtained as a result of the decoupling approximation. This property leads us to the Langevin

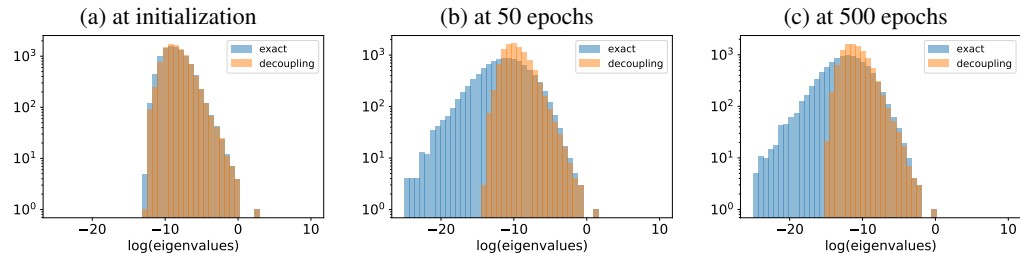

Figure 1: Comparison of the eigenvalue distributions of the left-hand side (exact expression) and the right-hand side (decoupled one) of Eq. (12) in the main text. They agree with each other except for very small eigenvalues during the entire training dynamics.

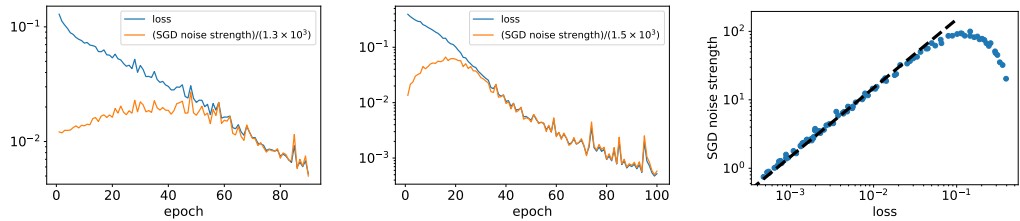

Figure 2: Training dynamics of the loss and the SGD noise strength $\mathcal{N}$. In the figure, we multiplied $\mathcal{N}$ by a numerical factor to emphasize that $\mathcal{N}$ is actually proportional to the loss in a later stage of the training. (Left) Fully-connected network trained by the Fashion-MNIST dataset. (Middle) Convolutional network trained by the CIFAR-10 dataset. (Right) the loss vs $\mathcal{N}$ in the training of the convolutional network. The dashed line is a straight line of slope 1, which implies $\mathcal{N} \propto L(\theta)$.

equation (14) with the logarithmized loss gradient and an additive noise through a random time change (13). Equation (14) implies the stationary distribution (17) and the escape rate (16).

In Sec. 5.1, we show that the decoupling approximation is valid during entire training dynamics. In Sec. 5.2, we measure the SGD noise strength and confirm that it is indeed proportional to the loss function near a minimum. In Sec. 5.3, we experimentally test the validity of Eq. (17) for the stationary distribution and Eq. (16) for the escape rate. We will see that numerical results for a linear regression and for a non-linear neural network agree with our theoretical results.

## 5.1 EXPERIMENTAL VERIFICATION OF THE DECOUPLING APPROXIMATION

Let us compare the eigenvalue distribution of the exact matrix $(1/N) \sum_{\mu=1}^{N} \ell^{(\mu)} C_f^{(\mu)}$ with that of the decoupled one $L(\theta) \cdot (1/N) \sum_{\mu=1}^{N} C_f^{(\mu)}$ with $C_f^{(\mu)} = \nabla f(\theta, x^{(\mu)}) \nabla f(\theta, x^{(\mu)})^{\mathrm{T}}$. We consider a binary classification problem using the first $10^4$ samples of the MNIST dataset such that we classify each image into even (its label is $y = +1$) or odd number (its label is $y = -1$). The network has two hidden layers, each of which has 100 units and the ReLU activation, followed by the output layer of a single unit with no activation. Starting from the Glorot initialization, the training is performed via SGD with the mean-square loss, where we fix $\eta = 0.01$ and $B = 100$.

Figure 1 shows histograms of their eigenvalues at different stages of the training: (a) at initialization, (b) after 50 epochs, and (c) after 500 epochs. We see that the exact matrix and the approximate one have statistically similar eigenvalue distributions except for very small eigenvalues during the training dynamics. This means that the decoupling approximation always holds during training.

## 5.2 MEASUREMENTS OF THE SGD NOISE STRENGTH

As a measure of the SGD noise strength, let us consider the norm of the noise vector $\xi$ given by $\langle \xi^{\mathrm{T}} \xi \rangle = \mathrm{Tr}\, \Sigma \approx \mathcal{N}/B$, where $\mathcal{N} := (1/N) \sum_{\mu=1}^{N} \nabla \ell_\mu^{\mathrm{T}} \nabla \ell_\mu - \nabla L^{\mathrm{T}} \nabla L$. Here we present experimental results for two architectures and datasets. First, we consider training of the Fashion-MNIST dataset by using a fully connected neural network with three hidden layers, each of which has $2 \times 10^3$ units and the ReLU activation, followed by the output layer of 10 units with no activation (classifica-

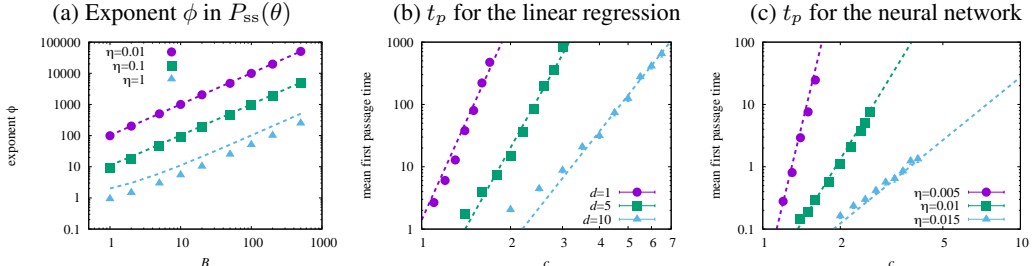

Figure 3: (a) Exponent $\phi$ for the stationary distribution $P_{\mathrm{ss}}(\theta) \propto L(\theta)^{-\phi}$ for $d = 1$ in the linear regression. Dashed lines show the theoretical prediction $\phi = 1 + B/(\eta h^*)$. (b) Log-log plot of the mean first-passage time $t_p$ vs $c = L(\theta^s)/L(\theta^*)$ for $B = 1$ and $\eta = 0.1$ in the linear regression. Dashed lines show the theoretical prediction, $t_p \propto \kappa^{-1} \propto c^{\phi - d/2}$. (c) Log-log plot of $t_p$ vs $c$ for $B = 10$ and various $\eta$ in the neural network. Dashed lines show the theoretical prediction, $t_p \propto \kappa^{-1} \propto c^{B/(\eta h^*) + 1 - n/2}$ with $h^* = 95.6$ and $n = 9$.

tion labels are given in the one-hot representation). Second, we consider training of the CIFAR-10 dataset by using a convolutional neural network. Following Keskar et al. (2017), let us denote a stack of $n$ convolutional layers of $a$ filters and a kernel size of $b \times c$ with the stride length of $d$ by $n \times [a, b, c, d]$. We use the configuration: $3 \times [64, 3, 3, 1]$, $3 \times [128, 3, 3, 1]$, $3 \times [256, 3, 3, 1]$, where a MaxPool(2) is applied after each stack. To all layers, the ReLU activation is applied. Finally, an output layer consists of 10 units with no activation.

Starting from the Glorot initialization, the network is trained by SGD of the mini-batch size $B = 100$ and $\eta = 0.1$ for the mean-square loss. During the training, we measure the training loss and the noise strength $\mathcal{N}$ for every epoch. Numerical results are given in Fig. 2. We see that roughly $\mathcal{N} \propto L$ at a later stage of the training, which agrees with our theoretical prediction.

Although $\mathcal{N}$ is not proportional to $L$ at an early stage of training, it does not mean that Eq. (9) is invalid there. Since the decoupling approximation is valid for the entire training dynamics, Eq. (9) always holds. The reason why the SGD noise strength does not decrease with the loss function in the early-stage dynamics is that $\mathcal{N} \approx 2L(\theta) \times (1/N) \sum_{\mu=1}^{N} \nabla f(\theta, x^{(\mu)})^{\mathrm{T}} \nabla f(\theta, x^{(\mu)})$, but the quantity $(1/N) \sum_{\mu=1}^{N} \nabla f(\theta, x^{(\mu)})^{\mathrm{T}} \nabla f(\theta, x^{(\mu)})$ also changes during training.

Although Eq. (9) is derived for the mean-square loss, the relation $\mathcal{N} \propto L(\theta)$ holds in more general loss functions; see Appendix D for general argument and experiments on the cross entropy loss.

## 5.3 Experimental Test of Stationary Distribution and Escape Rate Formula

We experimentally verify our theoretical predictions. Below, we first present numerical results for a simple linear regression problem, and then for a nonlinear model, i.e., a neural network.

Let us start from the following linear regression problem: each entry of $x^{(\mu)} \in \mathbb{R}^d$ and its label $y^{(\mu)} \in \mathbb{R}$ are i.i.d. Gaussian random variables of zero mean and unit variance. We focus on the case of $d \ll N$. The output for an input $x$ is given by $f(\theta, x) = \theta^{\mathrm{T}} x$, where $\theta \in \mathbb{R}^d$ is the trainable network parameter. We optimize $\theta$ via SGD. The mean-square loss $L(\theta) = (1/2N) \sum_{\mu=1}^{N} (\theta x^{(\mu)} - y^{(\mu)})^2$ is quadratic and has a unique minimum at $\theta \approx 0$.

First, we test Eq. (17), i.e. the stationary distribution, for $d = 1$ and $N = 10^5$. We sampled the value of $\theta_k$ at every 100 iterations ($k = j \times 100$, $j = 1, 2, \ldots, 10^4$) and made a histogram. We then fit the histogram to the form $P_{\mathrm{ss}}(\theta) \propto L(\theta)^{-\phi}$ and determine the exponent $\phi$. Our theory predicts $\phi = 1 + B/(\eta h^*)$. Numerical results for the exponent $\phi$ are presented in Fig. 3 (a) against $B$ for three fixed learning rates $\eta$. In the same figure, theoretical values of $\phi$ are plotted in dashed lines. The agreement between theory and experiment is fairly well. For a large learning rate $\eta = 1$, the exponent slightly deviates from its theoretical value. This is due to the effect of a finite learning rate (recall that $\eta$ is assumed to be small in deriving the continuous-time stochastic differential equation).

Next, we test our formula on the escape rate, Eq. (16). Although the mean-square loss is quadratic and no barrier crossing occurs, we can measure the first passage time, which imitates the escape time for a non-convex loss landscape. Let us fix a threshold value of the loss function. The first

passage time $t_p$ is defined as the shortest time at which the loss exceeds the threshold value. Here, time $t$ is identified as $\eta k$, where $k$ denotes the number of iterations in discrete SGD (2). We identify the threshold value as $L(\theta^s)$, i.e., the loss at the saddle in the escape problem. It is expected that $t_p$ is similar to the escape time and proportional to $\kappa^{-1}$.

The Hessian $H = (1/N) \sum_{\mu=1}^{N} x^{(\mu)} x^{(\mu)\mathrm{T}}$ has $d$ nonzero eigenvalues, all of which are close to unity. We can therefore identify $h^* = 1$ and $n = d$. The mean first passage time over 100 independent runs is measured for varying threshold values which are specified by $c = L(\theta^s)/L(\theta^*) > 1$. Experimental results for $N = 10^4$ are presented in Fig. 3 (b). Dashed straight lines have slope $B/(\eta h^*) + 1 - n/2$. Experiments show that the first passage time behaves as $t_p \propto [L(\theta^s)/L(\theta^*)]^{B/(\eta h^*)+1-n/2}$, which agrees with our theoretical evaluation of $\kappa^{-1}$ [see Eq. (16)]. We conclude that the escape rate crucially depends on the effective dimension $n$, which is not explained by the previous results (Zhu et al., 2019; Xie et al., 2021; Liu et al., 2021; Meng et al., 2020).

Our escape-rate formula (16) is also verified in a non-linear model. As in Sec. 5.1, we consider the binary classification problem using the first $10^4$ samples of MNINST dataset such that we classify each image into even or odd. The network has one hidden layer with 10 units activated by ReLU, followed by the output layer of a single unit with no activation. We always use the mean-square loss. Starting from thee Glorot initialization, the network is pre-trained via SGD with $\eta = 0.01$ and $B = 100$ for $10^5$ iterations. We find that after pre-training, the loss becomes almost stationary around at $L(\theta) \approx 0.035$. We regard that the pre-trained network is near a local minimum. We then further train the pre-trained network via SGD with a new choice of $\eta$ and $B$ (here we fix $B = 10$), and measure the first passage time $t_p$. The mean first-passage time over 100 independent runs is measured for varying threshold values which are specified by $c = L(\theta^s)/L(\theta^*) > 1$. Experimental results are presented in Fig. 3 (c). We see the power-law behavior, which is consistent with our theory.

To further verify our theoretical formula (16), we also compare the power-law exponent for the mean first-passage time with our theoretical prediction $B/(\eta h^*) + 1 - n/2$. Here, $h^*$ and $n$ are estimated by the Hessian eigenvalues. In Appendix E, we present a numerical result for eigenvalues of the approximate Hessian given by the right-hand side of Eq. (11). By identifying the largest eigenvalue of the Hessian as $h^*$, we have $h^* \approx 95.6$. On the other hand, it is difficult to precisely determine the effective dimension $n$, but it seems reasonable to estimate $n \sim 10$. It turns out that theory and experiment agree with each other by choosing $n = 9$. Dashed lines in Fig. 3 (c) correspond to our theoretical prediction $\kappa^{-1} \propto c^{B/(\eta h^*)+1-n/2}$ with $h^* = 95.6$ and $n = 9$. This excellent agreement shows that our theoretical formula (16) is also valid in non-linear models.

## 6 CONCLUSION

In this work, we have investigated SGD dynamics via a Langevin approach. With several approximations listed in Appendix A, we have derived Eq. (6), which shows that the SGD noise strength is proportional to the loss function. This SGD noise covariance structure yields the stochastic differential equation (14) with additive noise near a minimum via a random time change (13). The original multiplicative noise is reduced to simpler additive noise, but instead the gradient of the loss function is replaced by that of the logarithmized loss function $U(\theta) = \log L(\theta)$. This stochastic differential equation has a quasi-stationary distribution that decays polynomially with $L(\theta)$ near a minimum (17), not exponentially as in the usual Gibbs distribution. This new formalism yields the power-law escape rate formula (16) whose exponent depends on $\eta$, $B$, $h^*$, and $n$.

Our escape-rate formula explains an empirical fact that SGD favors flat minima with low effective dimensions. The effective dimension of a minimum must satisfy Eq. (18) for its stability. This result as well as the formulation of SGD dynamics using the logarithmized loss landscape should help understand more deeply the SGD dynamics and its implicit biases in machine learning problems.

Although the present work focuses on the Gaussian noise, the non-Gaussianity can also play an important role. For example, Şimşekli et al. (2019) approximated SGD as a Lévy-driven SDE, which explains why SGD finds wide minima. It would be an interesting future problem to take the non-Gaussian effect into account.

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

## A    LIST OF APPROXIMATIONS AND THEIR JUSTIFICATIONS

In Sec. 3.1, we made several approximations to derive the result (6). For clarity, we list the approximations made and their justifications below.

- In Eq. (8), we make the approximation of $B << N$, which simplifies the expression but is not essential. The main conclusion is not affected by this approximation.

- We ignore $\nabla L \nabla L^{\mathrm{T}}$ in Eq. (8), which is justified near a local minimum.

- In Eq. (9), we make the decoupling approximation (7), which is one of the key heuristic approximations in our work. This approximation is verified experimentally in Sec. 5.1.

- We ignore the last term of Eq. (10), which is a common approximation (Sagun et al., 2017). This approximation is justified if we are only interested in the outliers of the Hessian eigenvalues. Indeed, outliers of the Hessian eigenvalues, which play dominant roles in escape from local minima, are known to be attributable to the first term of the right-hand side of Eq. (10) (Papyan, 2018).

- In Eq. (11), we assume that the matrix $(1/N) \sum_{\mu=1}^{N} \nabla f(\theta, x^{(\mu)}) \nabla f(\theta, x^{(\mu)})^T$ does not change so much in a valley with a given local minimum. This approximation is indirectly verified in Fig. 2 (the proportionality between the loss and the noise strength implies this matrix is actually constant).

In deriving the escape rate for multi-variable cases in Sec. 4, we further make the following assumptions and approximations:

- First of all, we discard a bulk of near-zero eigenvalues of the Hessian and restricted learning dynamics within the subspace spanned by the outlier eigenvectors. This approximation is justified because the SGD dynamics is almost frozen along flat directions of the loss landscape.

- We approximate the SGD noise as an isotropic one within the $n$-dimensional outlier subspace. This approximation is justified either when the loss landscape in the outlier subspace is isotropic near the minimum or when the directions of the Hessian eigenvectors do not change within the valley.

- At the last part of the derivation, we approximate $\tau \approx L(\theta^*)t$, which is justified because the model stays near a local minimum until it escapes from that minimum.

## B    STATIONARY DISTRIBUTION

Since $\tilde{\theta}_\tau$ obeys a simple Langevin equation

$$d\tilde{\theta}_\tau = -U'(\tilde{\theta}_\tau) + \sqrt{2T}d\tilde{W}_\tau, \tag{20}$$

the stationary distribution of $\tilde{\theta}_\tau$ is given by the Gibbs distribution $\tilde{P}_{\mathrm{ss}}(\theta) \propto e^{-U(\theta)/T}$. On the other hand, what we want is the stationary distribution $P_{\mathrm{ss}}(\theta)$ of $\theta_t$, where $\theta_t = \tilde{\theta}_\tau$ with $\tau = \int_0^t dt'\, L(\theta_{t'})$. In this section, we show the relation between the two distributions: $P_{\mathrm{ss}}(\theta) \propto L(\theta)^{-1} \tilde{P}_{\mathrm{ss}}(\theta)$.

We express the stationary distributions in terms of the long-time average of the delta function:

$$P_{\mathrm{ss}}(\theta) = \lim_{s \to \infty} \frac{1}{s} \int_0^s dt\, \delta(\theta_t - \theta), \quad \tilde{P}_{\mathrm{ss}}(\theta) = \lim_{s \to \infty} \frac{1}{s} \int_0^s d\tau\, \delta(\tilde{\theta}_\tau - \theta). \tag{21}$$

By using the relation $\tau = \int_0^t dt'\, L(\theta_{t'})$, we have $d\tau = L(\theta_t)dt$. For a sufficiently large $t$, we also obtain $\tau \sim t\bar{L}$, where $\bar{L} := \lim_{s \to \infty}(1/s) \int_0^s dt'\, L(\theta_{t'})$ denotes the long-time average of $L(\theta_t)$. By

using them, $P_{\text{ss}}(\theta)$ is rewritten as

$$
\begin{aligned}
P_{\text{ss}}(\theta) &\approx \lim_{s\to\infty} \frac{1}{s} \int_0^{s\bar{L}} d\tau \, \frac{\delta(\tilde{\theta}_\tau - \theta)}{L(\tilde{\theta}_\tau)} \\
&= \frac{1}{L(\theta)} \lim_{s\to\infty} \frac{\bar{L}}{s\bar{L}} \int_0^{s\bar{L}} d\tau \, \delta(\tilde{\theta}_\tau - \theta) \\
&= \frac{\bar{L}}{L(\theta)} \tilde{P}_{\text{ss}}(\theta).
\end{aligned}
\tag{22}
$$

We thus obtain the desired relation, $P_{\text{ss}}(\theta) \propto L(\theta)^{-1} \tilde{P}_{\text{ss}}(\theta)$.

## C  JUSTIFICATION OF THE ISOTROPIC-NOISE APPROXIMATION WITHIN THE $n$-DIMENSIONAL SUBSPACE

We now show that the isotropic-noise approximation is valid when the directions of the eigenvectors of the Hessian do not change within the valley of a given local minimum. In this case, $\partial^2 U/\partial z_i \partial z_j = 0$ for any $i \neq j$, where the Hessian at the minimum is given by $H(\theta^*) \approx \sum_{i=1}^n h_i^* v_i v_i^{\mathrm{T}}$ ($h_i$ is an eigenvalue and $v_i$ is the corresponding eigenvector of $H(\theta^*)$) and the displacement vector $z \in \mathbb{R}^n$ is defined by $\theta = \theta^* + \sum_{i=1}^n z_i v_i$. The stochastic differential equation $dz = -\nabla_z U d\tau + \sqrt{2\eta H(\theta^*)/B} d\tilde{W}_\tau$ is then equivalent to the following Fokker-Planck equation for the distribution function $P(z,\tau)$ of $z$ at $\tau$:

$$
\frac{\partial P(z,\tau)}{\partial \tau} = \sum_{i=1}^n \left[ \frac{\partial}{\partial z_i} \left( \frac{\partial U}{\partial z_i} P \right) + \frac{\eta h_i^*}{B} \frac{\partial^2}{\partial z_i^2} P \right].
\tag{23}
$$

Let us assume that the direction of $e$th eigenvector $v_e$ corresponds to the escape direction. We denote by $z_\perp \in \mathbb{R}^{n-1}$ the displacement perpendicular to the escape direction, and write $z = (z_e, z_\perp)$. At the saddle $z^s$, $\nabla_z U(z^s) = 0$, $h_e^s < 0$ and $h_i^s > 0$ for all $i \neq e$, where $\{h_i^s\}$ is the set of eigenvectors of the Hessian at $z^s$.

Under the above setting, we derive the escape rate formula following Kramers (1940). The steady current $J \in \mathbb{R}^n$ is aligned to the escape direction, and hence $J_e \neq 0$ and $J_\perp = 0$. Let us denote by $P^*$ the total probability within the valley of a given minimum $\theta^*$ and by $\mathcal{J}$ the total current flowing to outside of the valley through the saddle $\theta^s = \theta^* + z^s$. It is assumed that $\mathcal{J}$ is small and $P^*$ is almost stationary. The escape rate $\kappa_\tau$ is then given by

$$
\kappa_\tau = \frac{\mathcal{J}}{P^*}.
\tag{24}
$$

The escape rate $\kappa$ per unit time is given by

$$
\kappa = L(\theta^*)\kappa_\tau = L(\theta^*)\frac{\mathcal{J}}{P^*}.
\tag{25}
$$

We now evaluate $P^*$ and $\mathcal{J}$. To evaluate $P^*$, it is necessary to know about the stationary distribution near the minimum $\theta^*$. Near the minimum $\theta^*$ (i.e. small $z$),

$$
U(\theta) = \log L(\theta) \approx U(\theta^*) + \frac{(\theta - \theta^*)^{\mathrm{T}} H(\theta^*)(\theta - \theta^*)}{2L(\theta^*)} = U(\theta^*) + \frac{z^{\mathrm{T}} H(\theta^*) z}{2L(\theta^*)}
\tag{26}
$$

and

$$
\frac{\partial U}{\partial z_i} \approx \frac{1}{L(\theta^*)} \sum_{j=1}^n H_{ij}^* z_j,
\tag{27}
$$

where $H^* = H(\theta^*)$. By substituting it into the Fokker-Planck equation (23), we obtain

$$
\frac{\partial P(z,\tau)}{\partial \tau} = \sum_{i=1}^n \frac{\partial}{\partial z_i} \left[ \sum_{j=1}^n H_{ij}^* \left( \frac{z_j}{L(\theta^*)} + \frac{\eta}{B} \frac{\partial}{\partial z_j} \right) P \right].
\tag{28}
$$

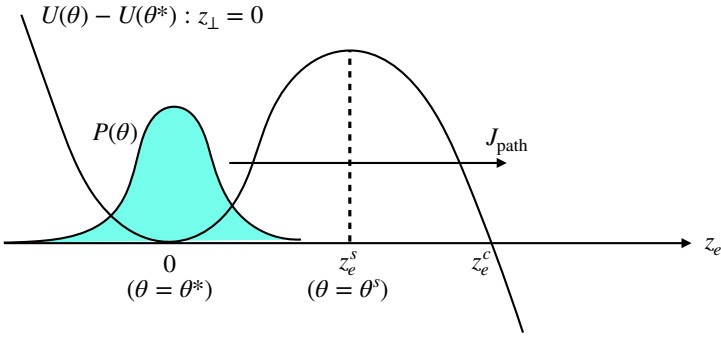

Figure 4: Schematic illustration of the escape from a potential barrier.

If for all $j$

$$\left(\frac{z_j}{L(\theta^*)} + \frac{\eta}{B}\frac{\partial}{\partial z_j}\right) P_{\text{ss}}(z) = 0, \tag{29}$$

$P_{\text{ss}}(z)$ is a stationary distribution near the minimum $\theta^*$. Indeed,

$$P_{\text{ss}}(z) = P(\theta^*)e^{-\frac{B}{2\eta L(\theta^*)}\sum_{i=1}^n z_i^2} \tag{30}$$

satisfies this condition. It should be noted that the stationary distribution is independent of the Hessian eigenvalues $\{h_i^*\}$. By using Eq. (30), $P^*$ is evaluated as

$$P^* \approx \int dz_1 dz_2 \dots dz_n P_{\text{ss}}(z) = P(\theta^*)\left[\frac{2\eta L(\theta^*)}{B}\right]^{n/2}. \tag{31}$$

Next, let us evaluate $\mathcal{J}$. The probability current $J$ along the escape direction $e$ is given by

$$J_e = -\frac{\partial U}{\partial z_e}P - \frac{\eta h_e^*}{B}\frac{\partial}{\partial z_e}P. \tag{32}$$

At the saddle, the current vector $J_\perp$ perpendicular to the escape direction is zero, and hence the probability distribution near the saddle with $z_e = z_e^s$ and $z_\perp \neq 0$ is also evaluated in a similar way as in Eq. (30):

$$P(z_e = z_e^s, z_\perp) \approx P(z_e = z_e^s, z_\perp = 0)e^{-\frac{B}{2\eta L(\theta^s)}z_\perp^2}. \tag{33}$$

By substituting it into Eq. (32), we obtain

$$J_e(z_e^s, z_\perp) = J_{\text{path}}e^{-\frac{B}{2\eta L(\theta^s)}z_\perp^2}, \tag{34}$$

where the current along the escape path ($z_\perp = 0$) is denoted by $J_{\text{path}} := J(z_e^s, z_\perp = 0)$. The total current through the saddle is then evaluated as

$$\mathcal{J} = \int dz_\perp J_e(z_e^s, z_\perp) = J_{\text{path}}\left[\frac{2\pi\eta L(\theta^s)}{B}\right]^{(n-1)/2}. \tag{35}$$

When the distribution function is almost stationary, Eq. (23) yields $\partial J_e(z_e, z_\perp = 0)/\partial z_e \approx 0$, and hence the current along the escape path is approximately constant $J_e(z_e, z_\perp = 0) \approx J_{\text{path}}$. Since $J_e(z_e, z_\perp = 0)$ is given by Eq. (32) by putting $z_\perp = 0$, we have

$$J_{\text{path}} = -\frac{\partial U}{\partial z_e}\bigg|_{z_\perp=0}P - \frac{\eta h_e^*}{B}\frac{\partial P}{\partial z_e} = -\frac{\eta h_e^*}{B}e^{-\frac{B}{\eta h_e^*}U}\frac{\partial}{\partial z_e}\left(e^{\frac{B}{\eta h_e^*}U}P\right). \tag{36}$$

By multiplying $e^{\frac{B}{\eta h_e^*}U}$ in both sides and integrating over $z_e$ from 0 to $z_e^c$, where $z_e^c$ defined as $U(\theta^* + z_e^c v_e) = U(\theta^*)$ (see Fig. 4), we obtain

$$J_{\text{path}}\int_0^{z_e^c} dz_e\, e^{\frac{B}{\eta h_e^*}U} = \frac{\eta h_e^*}{B}e^{\frac{B}{\eta h_e^*}U(\theta^*)}P(\theta^*), \tag{37}$$

where it is assumed that the probability at $z_e^c$ is small and negligible, $P(z_e = z_e^c, z_\perp = 0) \approx 0$. By using the saddle-point method, the integral in the left-hand side of Eq. (37) is evaluated as

$$
\int_0^{z_e^c} dz_e \, e^{\frac{B}{\eta h_e^*} U} \approx \int_{-\infty}^{\infty} dz_e \, \exp\left[ \frac{B}{\eta h_e^*} \left( U(\theta^s) + \frac{h_e^s}{2L(\theta^s)}(z_e - z_e^s)^2 \right) \right]
$$

$$
= \left( \frac{2\pi \eta h_e^*}{B|h_e^s|L(\theta^s)} \right)^{1/2} e^{\frac{B}{\eta h_e^*} U(\theta^s)}. \tag{38}
$$

By substituting this result in Eq. (37), we obtain

$$
J_{\text{path}} = \left( \frac{\eta h_e^* |h_e^s|}{2\pi B L(\theta^s)} \right)^{1/2} e^{-\frac{B}{\eta h_e^*} \Delta U} P(\theta^*), \tag{39}
$$

where $\Delta U = U(\theta^s) - U(\theta^*)$. The total current $\mathcal{J}$ in Eq. (35) is then expressed as

$$
\mathcal{J} = \frac{\sqrt{h_e^* |h_e^s|}}{2\pi L(\theta^s)} \left( \frac{2\pi \eta L(\theta^s)}{B} \right)^{n/2} e^{-\frac{B}{\eta h_e^*} \Delta U} P(\theta^*). \tag{40}
$$

By using Eqs. (31) and (40), the escape rate $\kappa$ in Eq. (25) is evaluated as

$$
\kappa = L(\theta^*) \frac{\mathcal{J}}{P^*} = \frac{\sqrt{h_e^* |h_e^s|}}{2\pi} \left[ \frac{L(\theta^s)}{L(\theta^*)} \right]^{\frac{n}{2}-1} e^{-\frac{B}{\eta h_e^*} \Delta U}. \tag{41}
$$

Since $\Delta U = \log[L(\theta^s)/L(\theta^*)]$, we finally obtain

$$
\kappa = \frac{\sqrt{h_e^* |h_e^s|}}{2\pi} \left[ \frac{L(\theta^s)}{L(\theta^*)} \right]^{-\left( \frac{B}{\eta h_e^*} + 1 - \frac{n}{2} \right)}, \tag{42}
$$

which is exactly identical to the escape rate formula derived in the main text using the isotropic-noise approximation with $h^* = h_e^*$.

In this way, the isotropic-noise approximation is justified even when the loss landscape is not isotropic near the minimum. We have assumed that the directions of the eigenvectors of the Hessian do not change within the valley of a given local minimum. Under this assumption, the Fokker-Planck equation along the escape path is decoupled from the perpendicular directions $z_\perp$. Moreover, we have seen that the stationary distribution near a minimum is independent of the Hessian eigenvalues. These properties explain the reason why the isotropic-noise approximation is justified in this case.

## D    OTHER LOSS FUNCTIONS

In our paper, we mainly focus on the mean-square loss, for which we can analytically derive the relation between the loss $L(\theta)$ and the SGD noise covariance $\Sigma(\theta)$. An important observation is that the SGD noise strength $\mathcal{N}$ is proportional to the loss, i.e., $\mathcal{N} \propto L(\theta)$ (see Sec. 5.2 of the main text for the definition of $\mathcal{N}$).

Here, we argue that the relation $\mathcal{N} \propto L(\theta)$ also holds in more general situations. During the training, the value of $\ell_\mu$ will fluctuate from sample to sample. At a certain time step of SGD, let us suppose that $N - M$ samples in the training dataset are already classified correctly and hence $\ell_\mu \sim 0$, whereas the other $M$ samples are not and hence $\ell_\mu \sim 1$. The loss function is then given by $L(\theta) = (1/N) \sum_{\mu=1}^{N} \ell_\mu \sim M/N$. When $\ell_\mu$ is small, $\nabla \ell_\mu$ will also be small. Therefore, for $N - M$ samples with $\ell_\mu \sim 0$, $\nabla \ell_\mu \sim 0$ also holds. The other $M$ samples will have non-small gradients: $\|\nabla \ell_\mu\|^2 \sim g$, where $g > 0$ is a certain positive constant. We thus estimate $\mathcal{N}$ as $\mathcal{N} \approx (1/N) \sum_{\mu=1}^{N} \nabla \ell_\mu^{\mathrm{T}} \nabla \ell_\mu \sim gM/N \sim gL(\theta)$. In this way, $\mathcal{N} \propto L(\theta)$ will hold, irrespective of the loss function.

However, we emphasize that this is a crude argument. In particular, the above argument will not hold near a global minimum because all the samples are correctly classified there, which implies that $\ell_\mu$ is small for all $\mu$, in contrast to the above argument relying on the existence of $M$ wrongly classified samples with $\ell_\mu \sim 1$ and $\|\nabla \ell_\mu\|^2 \sim g$.

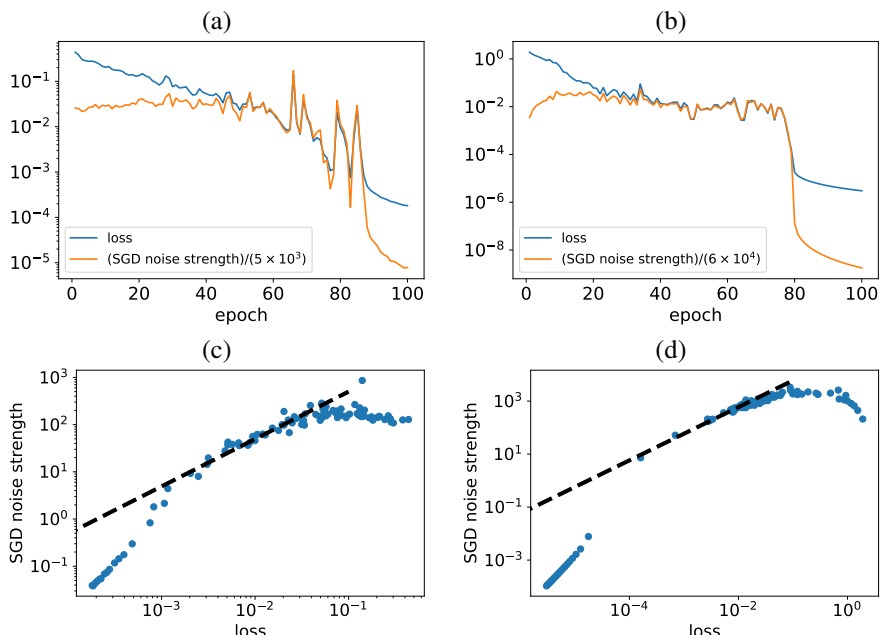

Figure 5: Training dynamics of the loss function and the SGD noise strength $\mathcal{N}$ for (a) a fully connected network trained by the Fashion-MNIST dataset and (b) a convolutional network trained by the CIFAR-10 dataset. In the figure, we multiplied $\mathcal{N}$ by a numerical factor to emphasize that $\mathcal{N}$ is actually proportional to the loss in an intermediate stage of the training. Loss vs $\mathcal{N}$ for (c) a fully connected network trained by the Fashion-MNIST and (d) a convolutional network trained by CIFAR-10. Dashed lines in (c) and (d) are straight lines of slope 1, which imply $\mathcal{N} \propto L(\theta)$.

We now experimentally test the relation $\mathcal{N} \propto L(\theta)$ for the cross-entropy loss. We consider the same architectures and datasets in Sec. 5.2 of the main text: a fully connected neural network trained by Fashion-MNIST and a convolutional neural network trained by CIFAR-10 (see Sec. 5.2 for the detail). We fix $B = 100$ in both cases, and $\eta = 0.1$ for the fully connected network and $\eta = 0.05$ for the convolutional network. Experimental results are presented in Fig. 5. We find that the relation $\mathcal{N} \propto L(\theta)$ seems to hold true at an intermediate stage of the training dynamics, although the proportionality is less clear compared with Fig. 2 in the main text for the mean-square loss.

We also find that for sufficiently small values of the loss, $\mathcal{N} \propto L(\theta)^2$ [see Fig. 5 (c) and (d)], whose implications should merit further investigation in future studies.

## E    HESSIAN EIGENVALUES FOR A NEURAL NETWORK IN SEC. 5.3

We present numerical results on Hessian eigenvalues in a pre-trained neural network studied in Sec. 5.3. Instead of the exact Hessian, we consider an approximate Hessian given on the right-hand side of Eq. (11), i.e.,

$$H(\theta^*) \approx \frac{1}{N} \sum_{\mu=1}^{N} \nabla f(\theta, x^{(\mu)}) \nabla f(\theta, x^{(\mu)})^{\mathrm{T}}. \tag{43}$$

Eigenvalues $\{h_i\}$ are arranged in descending order as $h_1 \geq h_2 \geq \cdots \geq h_P$ (in our model $P = 7861$).

A histogram of the Hessian eigenvalues is presented in Fig. 6 (a). We see that most eigenvalues are close to zero, which corresponds to the bulk, but there are some large eigenvalues, which correspond to the outliers. The largest eigenvalue is $\lambda_1 = 95.6$, which is identified as $h^*$ in our theoretical formula (16).

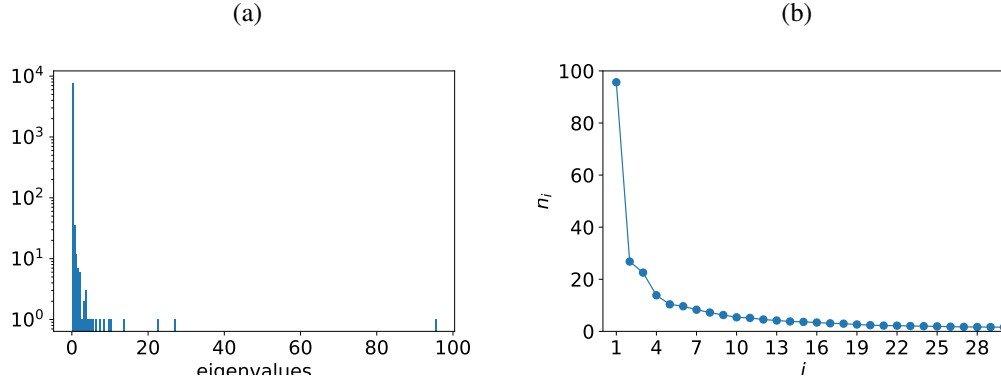

Figure 6: Hessian eigenvalues for a pre-trained neural network studied in Sec. 5.3. (a) The histogram of the Hessian eigenvalues. Most eigenvalues are close to zero, but there are some large eigenvalues, which correspond to outliers.

Another important quantity is the effective dimension $n$ corresnding to the number of outliers. Since the outliers and the bulk are not sharply separated, it is difficult to precisely determine $n$. In Fig. 6 (b), we plot $h_i$ up to $i = 30$. From this figure, it seems reasonable to estimate $n \sim 10$.

As a heuristic method of determining $n$, we can consider the following identification: first we define the weight $p_i$ for $i$th eigenmode as $p_i = h_i / \sum_{j=1}^{P} h_j$. We then determine $n$ as

$$n = \left( \sum_{i=1}^{P} p_i^2 \right)^{-1}, \tag{44}$$

which gives $n = 12.7$ in our case.

In Sec. 5.3, our formula (16) with $n = 9$ explains numerical results on the first-passage time.

