# OpenReview forum: "Logarithmic landscape and power-law escape rate of SGD"
_ICLR.cc/2022/Conference — ICLR 2022 Submitted_

### Official Review · Reviewer_zZQR · 2021-10-26

**Correctness:** 3
**Technical Novelty And Significance:** 3
**Empirical Novelty And Significance:** 2
**Recommendation:** 6
**Confidence:** 3

**Main Review:**

This paper studies the linear dependency of SGD noise with the loss. Theoretically, this linear dependency comes from a simplification of the noise covariance matrix. Numerically, it is verified for neural network models. With the linear dependency, SGD is formulated as a process with isotropic noise on the landscape of log-loss. Thus, the escape of SGD from minimum no longer depends exponentially on the barrier height. Rather, it depends polynomially on the height. Further, the escape rate depends on the effective non-vanishing dimension of the landscape. The connection of the stability of SGD around minima with the effective dimension is novel.

This paper is well written and well organized. Though the idea of studying the location-dependent, anisotropic noise of SGD is not new, this paper proposes a new approach to the problem by smartly simplify the noise covariance. The major concern of the reviewer is the lack of numerical support to the power law escape rate and stationary distribution for nonlinear models. Currently, models used in figure 3 is a linear regression. Is it possible to conduct similar experiments for nonlinear models such as neural networks? Will the results be different? I am willing to increase the score if strong numerical results are provided.

Some minor comments are:
1. In page 4, below equation (11), the authors mentioned "the last term in equation (11) does not contribute to outliers of Hessians". Could the authors provide some explanations? Also, it is good to explain what is an outlier of the Hessian. (though later we will know it means nonzero eigenvalues of the Hessian).

2. In page 6, above equation (20), it says "we assume anisotropy of the SGD noise within this n-dimensional space is not relevant". Why it is not relevant? The nonzero eigenvalues of Hessian can vary a lot, giving quite different noise strength along different directions. Could the authors provide justifications (either theoretically or numerically), that the behavior is not changed by considering an isotropic noise here.

**Summary Of The Paper:**

This paper studies the behavior of SGD around the minimum. Unlike many other works that simply treat SGD noise as a fixed noise, the authors characterizes the location-dependence of SGD noise, which gives drastically different escaping behavior. By some simplification of the noise covariance matrix, the authors are able to formulate an SGD with isotropic noise with a time change. The new SDE is a dynamics on the log-loss landscape with a simple additive noise. With this result, the escape rate and stationary distributions are derived, depending polynomially on the loss, instead of exponentially. Numerical experiments are conducted to justify the assumptions made in the analysis, and verify the theoretical conclusions in the case of linear regression.

**Summary Of The Review:**

This paper gives more realistic mathematical characterization of SGD's behavior around minima, by studying the location-dependence of SGD noise. The noise covariance is simplified in a smart way (with numerical justification) so that SGD dynamics can be formulated as a simple diffusion process on a log-loss landscape. Then, escape rate and stationary distribution are derived depending on a power law, instead of an exponential law.

---

> ### Author Response · Authors · 2021-11-15
> **Reply**
>
> We thank the reviewer for reading our paper and giving constructive comments. We reply to your comments and questions below.
>
> >The major concern of the reviewer is the lack of numerical support to the power law escape rate and stationary distribution for nonlinear models. Currently, models used in figure 3 is a linear regression. Is it possible to conduct similar experiments for nonlinear models such as neural networks? Will the results be different?
>
> Following your suggestion, we conducted experiments for a nonlinear model, i.e., a neural network trained by using the MNIST dataset. We have tested the validity of the escape-rate formula [Eq. (16) in the revised paper], and found that theory and experiment excellently agree with each other.
> Please see section 5.2 and see Figure 3 (c) in the revised manuscript.
>
> For nonlinear models, it is difficult to identify local minima and saddles. Therefore, we first find a point near a local minimum by pre-training of the model. The pre-trained model is used for initialization. Then, we measured the first passage time as in the linear regression problem. It is expected that the escape rate is proportional to the inverse of the first passage time, so we can test the validity of our formula. Numerical results presented in Figure 3 (c) show the power-law escape rate as predicted by our theory. Furthermore, we compare power-law exponents extracted by numerical results of the first passage time with the theoretical prediction. Since the theoretical exponent depends on $h^*$ and $n$, we have to estimate them. It is done by calculating eigenvalues of the Hessian (see Appendix E for the detail). By identifying the maximum Hessian eigenvalue as $h^*$ and $n=9$, we find that theory reproduces correct exponents.
>
>
> >In page 4, below equation (11), the authors mentioned "the last term in equation (11) does not contribute to outliers of Hessians". Could the authors provide some explanations? Also, it is good to explain what is an outlier of the Hessian. (though later we will know it means nonzero eigenvalues of the Hessian).
>
> It is nontrivial that the last term in Eq. (10) in the revised version does not contribute to outliers. We do not have intuitive explanations. Following your suggestion, we have inserted a remark that outliers mean large eigenvalues, just below Eq. (10) in the revised version.
>
> >In page 6, above equation (20), it says "we assume anisotropy of the SGD noise within this n-dimensional space is not relevant". Why it is not relevant? The nonzero eigenvalues of Hessian can vary a lot, giving quite different noise strength along different directions. Could the authors provide justifications (either theoretically or numerically), that the behavior is not changed by considering an isotropic noise here.
>
> As for the linear regression problem considered in section 5.3, the SGD noise is approximately isotropic. However, as you point out, outlier eigenvalues of the Hessian vary significantly in general, which is actually observed in the Hessian eigenvalues of a neural network presented in Appendix E.
>
> As we mention in the text, even if the Hessian at a local minimum is not isotropic, this assumption is justified when the directions of the Hessian eigenvectors do not change within the valley (see Appendix C for the details). We expect that this justification explains the fact that our theory agrees with experiments for the neural network model, although outlier eigenvalues of the Hessian vary a lot.

---

### Official Review · Reviewer_hfnj · 2021-11-02

**Correctness:** 4
**Technical Novelty And Significance:** 2
**Empirical Novelty And Significance:** Not applicable
**Recommendation:** 3
**Confidence:** 3

**Main Review:**

The paper gives an interesting characterization of the covariance matrix of SGD (note the noise due to data subsampling is structured) and uses a SDE persective, with the corresponding covariance to dervie power-law escape rates of SGD. The paper provides an intuitive introduction to the theoretical understanding of the papers and good sectional clarity in the way main concepts are presented.

However, the paper does not clearly define a lot of the variables used (such as the covariance matrix Sigma, which is a central quantity and deserves its own line) and much approximations are justified via empirical (and not mathematical) reasons. It's hard to understand what exactly is the novelty and main contribution of the paper, since many steps in the derivations are cited. It seems like the application of existing techniques on a restrictive class of functions (with mean-squared loss) may be somewhat interesting, but I'm unclear why (it's not clearly stated in the introduction).  Furthermore, the escape rate of SGD is with respect to a local minima, theta^*, so it is unclear how you define theta^* when it is not a global minima.



**Summary Of The Paper:**

The paper characterizes the escape rate of SGD for the mean-square loss via the perspective of SDE.

**Summary Of The Review:**

Due to the lack of novelty and the use of empirical justifications/lack of rigorous definitions, I would recommend a significant rewrite.

---

> ### Author Response · Authors · 2021-11-15
> **Reply**
>
> We thank the reviewer for reading our paper. We reply to your comments below.
>
> >However, the paper does not clearly define a lot of the variables used (such as the covariance matrix Sigma, which is a central quantity and deserves its own line) and much approximations are justified via empirical (and not mathematical) reasons.
>
> We are afraid that you overlooked the definitions of these variables. The covariance matrix $\Sigma$ is defined below Eq. (3), and its explicit form is given in Eq. (8). Although you mention that a lot of variables are not clearly defined, we fail to identify which ones they are. Certainly, a clear definition of the escape rate $\kappa$ was not given in the previous version, and hence we gave the definition at the beginning of section 4 in the revised manuscript. If there are other quantities which you think not clearly defined, could you please specifically point out them? It would help us to improve the paper.
>
> It is correct that some approximations are empirically justified. However, applied sciences and mathematics necessarily involve approximations, and justifying approximations with experiments is widely accepted as an acceptable and standard practice of science. The validity of the approximations is also supported by the fact that our theory agrees excellently with experiments as shown in section 5.
>
>
> >It's hard to understand what exactly is the novelty and main contribution of the paper, since many steps in the derivations are cited.
>
> As for the novelty and the main contribution of the paper, please see "Main contributions" and "Related works" on page 2. Technically, the decoupling approximation is a novel approximate method, by which it is shown that the SGD noise strength is proportional to the loss. The random time change in Eq. (13) in the revised paper is also a novel way to transform the original SDE with multiplicative noise to an SDE with much simpler additive noise. The main contribution of the paper is the power-law escape rate, i.e. Eq. (16), and its implication that SGD has biases towards flat minima with low effective dimension. In particular, the dependence on the effective dimension was not reported in previous works.
>
> We agree your assertion that "many steps in the derivations are cited", but we believe that it does not mean the lack of novelty. Any mathematical work necessarily rests on the results of previous works, and the novelty of the final result should be judged on its own and independent of how the results are derived. In addition, as we explained above, our paper indeed shows technical novelty (the decoupling approximation and the random time change). Therefore, your criticism cannot be scientifically justified.
>
> >It seems like the application of existing techniques on a restrictive class of functions (with mean-squared loss) may be somewhat interesting, but I'm unclear why (it's not clearly stated in the introduction).
>
> We fail to understand what the reviewer intends to say. Again, we emphasize that our theory, of course, relies on some previous studies, but some techniques used in our paper are novel. The main result, i.e. the power-law escape rate, is also novel on its own. We emphasize that an explanation on the novelty of our paper is clearly presented on page 2 of our paper.
>
> >Furthermore, the escape rate of SGD is with respect to a local minima, theta^*, so it is unclear how you define theta^* when it is not a global minima.
>
> A local minimum is defined as a point at which the gradient vanishes. This is a commonly accepted definition.

---

> > ### Comment · Reviewer_hfnj · 2021-11-22
> > **Author Response**
> >
> > Thanks for the response. I hope to clarify my criticisms, which are all in light that I believe that the paper's main contribution seems theoretical.
> >
> > 1) For the covariance matrix in defined below Eq (3) seems to be a random variable? Or is there an expectation over the randomness of the SGD.
> >
> > 2) Furthermore, it not clear how/where you can define escape rate of a optimization algorithm rigorously (is it in some region or at a point?) or what is the "basin of attraction" of a local minima (adding the definition of local minimum as a gradient-vanishing point will also be useful!).
> >
> > 3) There are no formal theorem statements in the paper, which I believe should be rectified with at least one formal and clear statement, so that others can cite your theoretical contribution.

---

> > > ### Author Response · Authors · 2021-11-22
> > > **Reply**
> > >
> > > Thank you for clarifying your comments and questions.
> > >
> > > >1. For the covariance matrix in defined below Eq (3) seems to be a random variable? Or is there an expectation over the randomness of the SGD.
> > >
> > > As we stated below Eq. (3), the brackets denote the average over possible choices of mini-batches (i.e. the average over the randomness of the SGD). So the covariance matrix is not a random variable but an expectation value.
> > >
> > > >2. Furthermore, it not clear how/where you can define escape rate of a optimization algorithm rigorously (is it in some region or at a point?) or what is the "basin of attraction" of a local minima (adding the definition of local minimum as a gradient-vanishing point will also be useful!).
> > >
> > > Thank you for your comments. In the revised manuscript, the definitions of the basin of attraction and the escape rate are given at the beginning of section 4.
> > >
> > > For convenience, let us repeat here. The basin of attraction $\mathcal{A}_{\theta^*}$ of a local minimum $\theta^*$ is defined as the set of all the initial points $\theta_0$ that arrive at the final state $\theta^*$ after a sufficiently long time evolution without noise.
> > >
> > > In order to exit from the basin of attraction, $\theta_t$ must overcome a saddle point $\theta^s$ with the help of noise. The escape rate is then defined as the inverse of the mean escape time from $\mathcal{A}_{\theta^*}$ starting from $\theta^*$. More formally,
> > >
> > > $\kappa=\langle \tau \rangle^{-1}$ with $\tau=\inf( t>0: \theta_t\notin\mathcal{A}_{\theta^*}, \theta_0=\theta^*)$.
> > >
> > > Following your suggestion, we added an explanation on a local minimum below Eq. (6) in the revised manuscript. At a local minimum, the gradient vanishes and the Hessian is positive semidefinite (i.e. every eigenvalue is non-negative).
> > >
> > > >3. There are no formal theorem statements in the paper, which I believe should be rectified with at least one formal and clear statement, so that others can cite your theoretical contribution.
> > >
> > > Our main results [equations (16) and (17)] are of approximate nature, and hence they are not mathematically rigorous. Therefore, we think that it is not suitable to present our results as theorem statements.
> > > Instead, by explicitly giving definitions of the basin of attraction and the escape rate at the beginning of section 4, we made an effort to make our statement clearer.

---

### Official Review · Reviewer_bWLZ · 2021-11-07

**Correctness:** 3
**Technical Novelty And Significance:** 4
**Empirical Novelty And Significance:** 3
**Recommendation:** 6
**Confidence:** 2

**Main Review:**

Strengths:
1. Overall, this paper is well written and easy to follow. The contribution of this paper is clearly presented.
2. To the best of my knowledge, the escape rate analysis of the SGD under the logarithmized loss landscape is novel as it partially explains why SGD prefers ﬂat minima with low effective dimensions.

Weakness:
1. There is a lack of clear definition of "escape rate". While I can somewhat guess the meaning of this quantity, I would prefer this critical concept to the explicitly and rigorously defined.
2. I also did not see a clear statement/reference of the escape rate (16). Maybe it is trivial to the authors, but I think for the broader audience, it would be very helpful if more reference/discuss could be included. I could not follow the discussion for the multidimensional case as well for the same reason.
3. In section 3.2, when the change of time scale is performed, are we still analyzing the (continuous time limit of) SGD iterates? I could not understand the meaning of the random time change from $t$ to $\tau$ as the loss $L$ is involved in (14). If we understand the variable $t$ as time, what is the variable $\tau$?

I think this paper is potentially important to the understanding of the bias of SGD, but I cannot fully comprehend the discussion in section 4.

**Summary Of The Paper:**

This paper considers the rate at which the SGD iterations will escape the valley around a local minimum. Under some approximation assumptions, this paper shows that the SGD noise covariance is highly structured as it aligns with the Hessian at the local minimum in the immediate vicinity. By considering the Ito SDE with the approximate SGD noise covariance and through a random change of time scale, the the Langevin equation on the loss landscape is transformed to that on the logarithmic loss landscape, but with a simpler additive noise. Such a logarithmized loss landscape is then exploited to derive the escape rate of SGD from a local minimum.

**Summary Of The Review:**

A potentially novel perspective for understanding the bias of SGD, but more discussion on the preliminary results are needed for non-expert readers to understand the derivation.

---

> ### Author Response · Authors · 2021-11-15
> **Reply**
>
> We thank the reviewer for carefully reading our paper and giving some useful comments. We reply to your comments below.
>
> >1. There is a lack of clear definition of "escape rate". While I can somewhat guess the meaning of this quantity, I would prefer this critical concept to the explicitly and rigorously defined.
>
> Thank you for your comment. At the beginning of section 4 in the revised manuscript, we clearly define the escape rate.
>
>
> >2. I also did not see a clear statement/reference of the escape rate (16). Maybe it is trivial to the authors, but I think for the broader audience, it would be very helpful if more reference/discuss could be included. I could not follow the discussion for the multidimensional case as well for the same reason.
>
> We noticed that it was hard to understand the organization of section 4 in the previous version, sorry for that.
> In section 4 of the previous manuscript, we first present the main results [Eqs. (15), (16), and (17)] without derivations. Their derivations are given later (on page 6), in which we listed some relevant references.
>
> In the revised manuscript, we have added a paragraph at the beginning of section 4, which contains the definition of the escape rate and our method of evaluating it [i.e. the use of the Kramers formula for Eq. (14)]. We hope that this additional explanation helps readers to understand the idea behind our main results.
>
> Before stating Eq. (15), we also declare "First, we present main results [...], and give their derivations later".
> Equation (15) is simply obtained by applying the Kramers formula to Eq. (14), so we also add a reference just below Eq. (15).
> We hope that this revision makes the paper more understandable.
>
> >3. In section 3.2, when the change of time scale is performed, are we still analyzing the (continuous time limit of) SGD iterates? I could not understand the meaning of the random time change from $t$ to $\tau$ as the loss $L$ is involved in (14). If we understand the variable $t$ as time, what is the variable $\tau$?
>
> Thanks for this question. The random time change should be regarded as a theoretical trick to make the problem simpler. By changing the time scale from $dt$ to $d\tau$, the noise strength apparently changes. By using this property, the local time scale $d\tau(\theta)$ at $\theta$ is introduced so that the noise strength becomes uniform. It is done by choosing $d\tau(\theta)=L(\theta)dt$ as done in our paper.
> The new variable $\tau$ is an accumulated value of this local time difference, i.e., $\tau=\int d\tau=\int L(\theta_t)dt$.
>
> >I think this paper is potentially important to the understanding of the bias of SGD, but I cannot fully comprehend the discussion in section 4.
>
> We appreciate your acknowledgement of the potential importance of our paper. We hope that the above reply clarifies the points you raised.

---

> > ### Comment · Reviewer_bWLZ · 2021-11-29
> > **Response to the reply**
> >
> > I thank the authors for clarifying the concept of escape rate. However, I am still not convinced that after the change of time scale, we are still analyzing the continuous time limit of SGD iterates. Hence my score remains unchanged.

---

> > > ### Author Response · Authors · 2021-11-29
> > > **relation between $\tau$ and $t$**
> > >
> > > Thank you for your feedback. We would like to make the point clearer.
> > >
> > > >However, I am still not convinced that after the change of time scale, we are still analyzing the continuous time limit of SGD iterates.
> > >
> > > Our primary interest is the escape from a local minimum $\theta^*$. Since the value of the loss function is close to $L(\theta^*)$ before escape, $\tau$ is approximately equal to $L(\theta^*)t$. Therefore, $\tau$ is identical to $t$ up to a scaling factor. It means that we are analyzing the continuous time limit of SGD iterates even after the random time change.
> > >
> > > That is why, we can derive the escape rate formula, i.e. eq. (16), for the continuous time limit of SGD iterates by using the random time change. Please see the discussion below eq. (19).

---

### Author Response · Authors · 2021-11-15
**Rebuttal Summary**

We thank the reviewers for reading our manuscript and giving constructive comments/questions. We reply to each question and comment of the reviewers below.

Summary of the revision:
* We added an experimental result for the escape rate in Figure 3 (c). We find an excellent agreement between theory and experiment. Details are explained in section 5.3.
* We added Appendix E, in which we present numerical results for the (approximate) Hessian eigenvalues for a neural network model discussed in section 5.3.
* At the beginning of section 4, we added a paragraph in which we define the escape rate and briefly explain the method of how we get formulae on the escape rate.
* Some minor modifications to satisfy the page limit.

---

> ### Author Response · Authors · 2021-11-22
> **Typo fixed**
>
> There was a typo in the definition of the escape time at the beginning of section 4.1.
> Crucially,
> $\theta_t\in\mathcal{A}_{\theta^*}$
>
> in the definition of $\tau$ should be
>
> $\theta_t\notin\mathcal{A}_{\theta^*}$.
>
> We fixed it in the revised manuscript.

---

### Decision · Program_Chairs · 2022-01-20

**Decision:**

Reject

**Comment:**

The paper studies rate at which SGD escapes local minima and provides a potential justification for the "flat minima" observation.
Reviewers agree that the paper studies an interesting problem and provides a nice result. But it seems like that paper in it's current shape is not ready for publication at ICLR. Issue is that the paper's writing is not up to bar, and requires a fair bit of work. In particular, the paper doesn't define the key quantities formally, doesn't provide all the assumptions in one place and justify why they might be reasonable. Finally, it would be great if the final result about escape rate is provided clearly with a self contained theorem/lemma that define/describe most of the key quantities in the rate.